# Application of Convolutional Neural Networks and Recurrent Neural Networks in Food Safety

**DOI:** 10.3390/foods14020247

**Published:** 2025-01-14

**Authors:** Haohan Ding, Haoke Hou, Long Wang, Xiaohui Cui, Wei Yu, David I. Wilson

**Affiliations:** 1Science Center for Future Foods, Jiangnan University, Wuxi 214122, China; dinghaohan@jiangnan.edu.cn; 2School of Artificial Intelligence and Computer Science, Jiangnan University, Wuxi 214122, China; 6243110027@stu.jiangnan.edu.cn (H.H.); 6233115022@stu.jiangnan.edu.cn (L.W.); 3School of Cyber Science and Engineering, Wuhan University, Wuhan 430072, China; 4Department of Chemical & Materials Engineering, University of Auckland, Auckland 1010, New Zealand; w.yu@auckland.ac.nz; 5Electrical and Electronic Engineering Department, Auckland University of Technology, Auckland 1010, New Zealand; diwilson@aut.ac.nz

**Keywords:** food safety, deep learning, convolutional neural networks, recurrent neural networks, long short-term memory

## Abstract

This review explores the application of convolutional neural networks (CNNs) and recurrent neural networks (RNNs) in food safety detection and risk prediction. This paper highlights the advantages of CNNs in image processing and feature recognition, as well as the powerful capabilities of RNNs (especially their variant LSTM) in time series data modeling. This paper also makes a comparative analysis in many aspects: Firstly, the advantages and disadvantages of traditional food safety detection and risk prediction methods are compared with deep learning technologies such as CNNs and RNNs. Secondly, the similarities and differences between CNNs and fully connected neural networks in processing image data are analyzed. Furthermore, the advantages and disadvantages of RNNs and traditional statistical modeling methods in processing time series data are discussed. Finally, the application directions of CNNs in food safety detection and RNNs in food safety risk prediction are compared. This paper also discusses combining these deep learning models with technologies such as the Internet of Things (IoT), blockchain, and federated learning to improve the accuracy and efficiency of food safety detection and risk warning. Finally, this paper mentions the limitations of RNNs and CNNs in the field of food safety, as well as the challenges in the interpretability of the model, and suggests the use of interpretable artificial intelligence (XAI) technology to improve the transparency of the model.

## 1. Introduction

In the field of international public health and the food industry, food safety represents a critical issue that directly impacts public health and social stability. In recent years, the frequent occurrence of food safety incidents, including biochemical contamination, food adulteration, and the sale of expired products, has not only posed significant threats to consumer health but also led to substantial economic losses and social unrest [1]. According to statistics from the World Health Organization (WHO), approximately 600 million people worldwide fall ill each year due to foodborne diseases, resulting in 420,000 deaths [2]. The efficient detection of food safety issues and the prediction of associated risks, coupled with strengthened regulatory measures, are not only fundamental requirements for safeguarding consumer rights but also pivotal strategies for promoting the sustainable development of the food industry.

Traditional methods for food safety testing encompass a variety of techniques, primarily including physical detection (e.g., standard sensors [3]), chemical analysis (e.g., high-performance liquid chromatography [4] and mass spectrometry [5]), and biological detection technologies (e.g., culture-based methods and biosensors [6]). Some of these technologies have reached a relatively mature stage of development. For instance, standard sensors are widely used for routine monitoring in food processing environments due to their ability to provide real-time online monitoring of critical parameters such as temperature, humidity, and pressure [7]. High-performance liquid chromatography (HPLC), capable of detecting trace chemical components in food (e.g., pesticide residues and illegal additives), has demonstrated notable cost-effectiveness and relatively high analysis speed in offline testing scenarios [8]. However, traditional detection methods exhibit certain limitations. Physical detection methods are primarily confined to environmental parameters and lack sufficient capability for analyzing food composition [9]. Chemical analysis techniques, such as HPLC, often require laboratory environments, making them unsuitable for large-scale online testing within the supply chain. Additionally, these methods heavily rely on standard calibration, making them prone to analytical biases when operating outside standard conditions [4]. As the diversity of food products and safety risks continues to grow, and as detection tasks increasingly shift toward real-time and high-throughput demands, the shortcomings of traditional methods in processing multi-modal data and analyzing dynamic changes are becoming increasingly evident.

Traditional food safety risk prediction methods mainly include statistical modeling methods and traditional machine learning models, which show certain application value in specific scenarios. Methods based on statistical modeling are usually suitable for situations with small data volumes and relatively simple features [10,11]. For example, regression analysis performs simple prediction tasks through linear relationships [12]. Time series analysis uses historical data to predict future risks and trends [13]. With the combination of prior probability and conditional dependence, the Bayesian method can effectively update the prediction result gradually when sufficient historical data are accumulated [14]. However, these statistical modeling methods show obvious limitations when dealing with complex nonlinear relationships, long-term dependencies, and dynamically changing large-scale data. Traditional machine learning models often rely on large datasets to achieve adequate pattern learning and accurate risk prediction [15]. For example, decision trees, with their hierarchical decision structures, can be applied to food attribute analysis and pollution risk prediction, but the generalization ability of the model is weak. By using kernel functions, support vector machines (SVMs) can effectively capture nonlinear relationships in data and perform well in high-dimensional data environments [16]. However, the computational complexity on large-scale datasets is high, and the training efficiency is low. Although these traditional machine learning models can achieve acceptable predictive accuracy in many cases, they generally rely on manual feature engineering, and it is difficult for them to automatically learn complex feature patterns in data. In addition, the performance of these models is often limited when faced with increasing data dimensions, dynamically changing data environments, and long time series dependencies. Especially in the case of large data noise and uneven distribution of categories, its prediction performance will be affected.

Deep learning, as a significant branch of artificial intelligence, has demonstrated tremendous potential in handling complex data, enabling automated analysis, and achieving efficient predictions, gradually becoming a crucial tool in food safety detection [17]. Compared to traditional food safety detection and risk prediction methods, deep learning technology leverages its powerful data mining capabilities to not only enhance speed and efficiency but also automatically extract key features from massive datasets, enabling more accurate detection of food safety issues and risk prediction [15,18]. Among deep learning architectures, convolutional neural networks (CNNs) and recurrent neural networks (RNNs) are two prominent models, each excelling in handling different types of data.

CNNs exhibit outstanding performance in processing image data and extracting image features, making them highly valuable and promising in food safety detection within the food industry [19]. CNNs can efficiently extract multi-level features such as color, shape, and texture from food images, making them applicable in scenarios like food appearance inspection, surface defect recognition, and adulteration analysis [20]. For instance, CNNs can rapidly identify decayed areas by analyzing the surface texture of fruits or automatically detect defective products on food packaging lines. Moreover, CNNs combined with super-resolution imaging technology can precisely analyze microscopic contaminants, significantly enhancing the reliability of food safety detection [21].

RNNs excel in handling time series data, making them particularly important for risk forecasting in food production and distribution. RNNs are adept at capturing temporal correlations in dynamic processes, making them more suitable for predictive time series modeling [16]. For example, by analyzing temporal changes in processing parameters such as temperature and humidity, RNNs can predict potential contamination risks in food production environments [22]. In cold chain logistics, RNNs can monitor transportation conditions in real time to prevent food spoilage. Particularly, the RNN variant known as Long Short-Term Memory (LSTM) networks effectively capture long-term dependencies, making them ideal for complex food safety data prediction tasks [23].

In food safety, the combined application of CNNs and RNNs allows each to leverage its technical advantages in image detection and time series prediction, forming a complementary deep learning framework [24]. The multi-feature recognition capabilities of CNNs enable efficient detection of food components and appearance, such as raw material sorting, spoilage inspection, and defect screening. Meanwhile, the time series prediction capabilities of RNNs excel in dynamic risk warning during food processing. In the food supply chain, the collaboration between CNNs and RNNs not only improves operational efficiency but also ensures food safety by utilizing their respective strengths [25]. For instance, in the raw material procurement stage, CNNs can analyze image data provided by suppliers to identify pest infestations, assess ripeness, and other quality indicators, while RNNs can analyze suppliers’ historical data to predict the stability and reliability of their supply [26]. In food processing and production lines, CNNs can extract features from food images, which, when combined with RNNs to analyze patterns of these features over time, enable real-time detection of food appearance and comprehensive analysis of food composition. This synergy ensures safety throughout the entire production process.

This review focuses on the research and literature achievements of convolutional neural networks (CNNs) and recurrent neural networks (RNNs) in the field of food safety. The main objective is to systematically discuss and summarize the specific application, and technical advantages and limitations of CNNs and RNNs in food safety detection and risk prediction, and provide valuable reference and guidance for future research direction and practical application. Figure 1 provides an overview of the different application areas of CNNs and RNNs in food safety: CNNs are mainly used for food safety inspection, which is further subdivided into six specific directions. RNNs are more suitable for food safety risk warning and are subdivided into four specific directions. This figure provides an overall overview of the contents of the third and fourth parts of this paper.

The structure of this paper is organized as follows: Section 2 compares CNNs with traditional neural networks and explores the differences between RNNs and traditional statistical modeling methods, highlighting the unique advantages of CNNs in image recognition tasks and the superiority of RNNs and their variants (e.g., LSTM) in handling time series data. Section 3 and Section 4 summarize the current applications of CNNs in food safety detection and the research progress of RNNs and LSTMs in food safety risk prediction from different perspectives. These sections also provide a comparative analysis of their technical advantages, limitations, and future development directions. Section 5 discusses the prospects of deep learning technologies in the future of food safety, particularly their integration with cutting-edge technologies such as the Internet of Things (IoT) and blockchain, as well as the challenges related to interpretability in practical applications.

## 2. Principles and Architectures of CNN and RNN Technologies

This section summarizes the core ideas and architectures of CNNs and RNNs, and describes the advantages and limitations of the advanced architectures of CNNs (such as AlexNet, VGGNet, etc.) and the important variants of RNNs (LSTM, GRU). By comparing CNNs with traditional fully connected neural networks, the advantages of CNNs for image feature recognition and extraction are highlighted. At the same time, RNNs are more applicable and robust than feedforward neural networks and traditional statistical modeling methods for processing time series data.

### 2.1. Principles and Architecture of CNNs

Convolutional neural networks (CNNs) are particularly adept at processing data with a two-dimensional structure, such as image data. The core idea of CNNs lies in extracting local features through convolution operations and learning complex patterns via multi-level feature representations [27]. Compared to fully connected neural networks, the convolution operations in CNNs allow for more effective capture of local correlations in images while avoiding the excessive number of parameters required by fully connected layers [28]. For instance, processing a 1000 × 1000 pixel image with fully connected neural networks would involve millions of parameters, whereas CNNs significantly reduce the computational load through convolution and pooling operations, making training more efficient. Furthermore, CNNs leverage their deep network architecture to extract features ranging from low-level (e.g., edges and textures) to high-level (e.g., shapes and structures), a capability that is challenging for FCNNs to achieve.

The architecture of a convolutional neural network (CNN) primarily consists of an input layer, convolutional layers, activation functions, pooling layers, fully connected layers, and an output layer [29]. The convolutional layer uses kernels sliding over the input data to extract local features, enhancing the model’s nonlinearity through activation functions. The pooling layer reduces the spatial dimensions of feature maps, decreasing computational complexity and mitigating overfitting. The fully connected layer flattens the dimensionality-reduced feature maps into vectors and computes weighted outputs for classification. The output layer typically employs functions like softmax or Sigmoid to generate the final prediction results [30]. Regularization techniques, such as Dropout, are often applied to prevent overfitting [31]. Through a sequence of convolution, pooling, activation, and fully connected operations, the network gradually extracts high-level features from the input data and maps them to the output space, enabling tasks such as classification, regression, or other applications [32]. Figure 2 shows the basic architecture of CNNs, illustrating its working principle through the process of food type recognition. The input layer receives an image of a fruit, and the convolution layer extracts local features of the image, such as shape and edge information, through multiple convolution cores. The pooling layer reduces dimensions, preserves critical information, and improves computational efficiency. The fully connected layer then integrates the compressed image features and correlates them with different categories of food, and the output layer finally gives the result of food recognition and the probability of each category, such as 89% for oranges, 10% for lemons, and 1% for mangoes in the figure.

On the foundation of the standard CNN architecture, many advanced variants have been proposed to enhance the model’s representational capacity, computational efficiency, and adaptability to various tasks [33]. AlexNet introduced a deeper network structure and larger convolutional kernels, replacing the Sigmoid activation function in standard CNNs with ReLU, enabling more detailed feature extraction [34]. VGGNet employed small convolutional kernels (3 × 3) stacked across multiple layers, achieving a more modular and efficient network design, which delivered exceptional performance in image classification tasks [35]. ResNet introduced residual connections, addressing the vanishing gradient problem in deep neural networks and allowing the network depth to extend to hundreds or even thousands of layers. InceptionNet utilized multiple convolutional kernel sizes (e.g., 1 × 1, 3 × 3, 5 × 5) within the same layer, enabling multi-scale feature extraction, which is suitable for analyzing diverse image features, such as fine surface textures and overall shapes of food [36]. DenseNet implemented dense inter-layer connections, wherein each layer utilized the feature outputs of all preceding layers, promoting feature reuse. This characteristic made DenseNet particularly effective in scenarios with limited data [37]. Additionally, the Vision Transformer (ViT), which has demonstrated excellent performance in image processing, employs self-attention mechanisms to capture global relationships across different regions of an image. When combined with CNNs, it offers enhanced global feature capture capabilities, significantly improving image detection accuracy [38]. While improving the performance of each CNN variant model, there are also limitations. AlexNet has high hardware requirements and a shallow network that is not suitable for complex image processing tasks. VGGNet is difficult to deploy in resource-constrained scenarios due to the large number of parameters and computing costs. ResNet networks introduce redundant structures as the number of layers increases, reducing computational efficiency. However, the network structure of InceptionNet and DenseNet is complex, and although it is suitable for complex image analysis, it is also prone to problems such as difficult parameter adjustment and weak scalability. These models need to weigh options based on application requirements.

### 2.2. Principles and Architecture of RNNs

Recurrent neural networks (RNNs) are deep learning models well suited for processing time series data. Their core characteristic lies in the recurrently connected neurons that pass information across time steps, enabling the model to capture contextual relationships within sequential data [39]. Unlike traditional feedforward neural networks, which lack memory and treat each input independently, RNNs’ recurrent mechanism allows the networks’ current output to be influenced by previous inputs, thereby enabling time series modeling [40]. Traditional statistical models, such as ARIMA and VAR, are also commonly used for system identification and predictive modeling of time series data. However, these methods rely heavily on assumptions such as the stationarity of the time series and the linear relationships within the data [41]. While these assumptions perform well in simple, short-cycle time series problems, they often result in significant prediction errors when faced with complex, nonlinear, high-dimensional, or long-term dependent time series data. For example, in food processing, the interrelationships between variables like temperature, humidity, and pressure are often highly nonlinear [42]. RNNs can capture the complex nonlinear dynamics in time series data without relying on linear assumptions, overcoming the limitations of traditional statistical methods and providing more accurate dynamic risk predictions.

The basic architecture of a recurrent neural network (RNN) includes an input layer, a hidden layer, and an output layer. The input layer receives the input at each time step of the sequential data, typically represented as a vector [43]. The hidden layer consists of one or more hidden units connected in a recurrent structure. At each time step, the hidden state is determined by the current input and the hidden state from the previous time step, allowing the network to retain and utilize historical information [44]. The output layer can take different forms depending on the task requirements. For classification tasks, the output layer is usually a fully connected layer followed by a softmax function, which outputs the probabilities for each class. For sequence generation tasks, the output layer generates predictions for the next time step. In RNNs, the weights for each time step are shared across the entire sequence, meaning the network uses the same parameters for different time step inputs. This weight-sharing mechanism significantly reduces the number of parameters in the model and enhances its generalization capability [45]. Figure 3 presents the basic architecture of an RNN in both abstract and detailed forms, illustrating how the hidden state from the previous time step influences the current hidden state.

However, standard RNNs often encounter the problem of vanishing or exploding gradients when processing long sequences [43]. To address the vanishing gradient issue in long-sequence modeling, Long Short-Term Memory (LSTMs) networks were developed. LSTMs introduce a gating mechanism, including input gates, forget gates, and output gates, which effectively controls the flow of information across time steps, enabling the modeling of long-term dependencies. This design allows the model to selectively retain or forget information, thus mitigating the vanishing gradient problem [46]. In contrast, another RNN variant, the Gated Recurrent Unit (GRU), simplifies the gating structure, further reducing the computational complexity of the model [47]. Figure 4 illustrates the internal structure of an LSTM, providing a clear depiction of how an LSTM neural network unit functions.

## 3. Food Safety Detection Using CNNs

This part systematically summarizes the research status and application progress of CNN technology in the field of food safety inspection. Firstly, the technical advantages of CNN in food image processing and feature extraction are analyzed, and the important position of food safety detection in food supply chain is expounded. Then, the application of CNN in food safety testing is subdivided into six directions in the form of the table, including pollution detection, fraud detection, etc., and its application status is summarized. On this basis, different indicators were used to compare the application directions, combined with critical analysis and advantages, and further revealed the potential and shortcomings of CNN technology in food safety detection. Finally, the combined application of CNN and Vision Transformer (ViT) in food safety detection in recent years is summarized, emphasizing the synergistic advantages of the two in feature expression and detection performance as well as the limitations of integration, which provides a reference direction for future research and practice.

### 3.1. Current Applications of CNNs

Food safety detection plays a critical role throughout the entire food supply chain, ensuring strict adherence to hygiene and safety standards at every stage from farm to table to prevent factors that could endanger consumer health [48]. This includes not only detecting harmful substances or contamination in food but also evaluating freshness, nutritional content, and the presence of adulteration or substandard processing. Additionally, food detection involves tasks such as food identification and classification to ensure product compliance with quality standards and market demands. Essentially, the core objective of food safety detection is to determine whether food meets established quality standards and to scientifically assess its safety [49]. In the context of an increasingly complex global food supply chain, food safety detection must not only achieve high precision but also ensure efficiency, real-time capabilities, and adaptability to diverse scenarios [50]. Figure 5 shows the workflow of food safety detection based on CNN, indicating that different indicators should be used in different scenarios to evaluate whether there are problems with food.

In food safety testing, image data usually involve details such as the surface, internal defects, foreign bodies, and contaminants of the food, and this information is closely related in space. CNNs are able to automatically extract these local features, such as cracks, stains, and mildew on food surfaces, and effectively maintain the spatial structure of the image through convolution operations [20]. This feature makes CNNs superior to traditional fully connected neural networks (FCNNs) in extracting spatial features from food-related images. The convolution operation of CNNs allows the networks to detect local features in a translation invariant way, regardless of where they appear in the image, which is important when processing food images, as defects or contaminants can appear in any part of the image. Moreover, for color images, convolution operations can be performed simultaneously on multiple color channels, ensuring that complete color information is captured, which is important for detecting color changes in food products, such as signs of spoilage [51]. In contrast, the input layer of FCNN requires the two-dimensional image to be flattened into a one-dimensional vector, ignoring the spatial relationship between pixels, and it is difficult to process food images with high complexity. For example, in original food images, adjacent pixels often have similar features or form a certain pattern (such as edges, textures, etc.), but after flattening, these spatial dependencies are broken, resulting in the model being unable to capture local features effectively. On the other hand, FCNN also relies on manually designed feature engineering, which sometimes requires experts to manually select and extract features. By automatically learning features, CNN simplifies this process and improves the accuracy and efficiency of detection [52]. Table 1, Table 2, Table 3, Table 4, Table 5 and Table 6 summarize the current research status of CNN in food safety detection, covering six key directions: contamination detection, ingredient analysis, pesticide residue detection, adulteration detection, food classification and identification, and damage detection. The tables provide a multi-dimensional overview of target tasks, technical methods, equipment usage, data sources, accuracy performance, and technological maturity. It highlights the commonalities and differences across application areas and demonstrates the broad potential of CNN in image processing and feature recognition for food safety detection.

By comparing the commonalities and differences across various application directions under different metrics, the tables provide a relatively clear and reasonable presentation of the current research status and potential value of CNN technology in food safety detection. Firstly, in terms of technical methods, CNN models in food safety detection are often combined with other technologies to enhance applicability and performance. The integration of hyperspectral imaging (HSI) with CNN is a major feature of current technologies, while tasks involving RGB images are more reliant on CNN’s standard feature extraction capabilities. Most food composition analysis and contamination detection tasks combine HSI to supplement the visual features with high-dimensional spectral data. For the linear characteristics of spectral data, 1D-CNN is also widely used for feature extraction and dimensionality reduction. In areas such as pesticide residue and contamination detection, traditional methods (e.g., PLS-DA, random forests) are often employed for comparative analysis of model performance. Adulteration detection and classification tasks, on the other hand, are partially based on RGB images, where CNN directly processes visual features without the need for complex spectral preprocessing. Secondly, regarding dataset sources, the primary limitation in current research is the insufficient scale and diversity of datasets. The datasets used in experiments are mostly collected in laboratory settings, with limited scale, typically ranging from hundreds to thousands of samples. Moreover, the singularity of data sources and distributions (e.g., standard laboratory conditions) restricts the model’s ability to generalize to the complexity of real-world scenarios. In terms of accuracy, the detection accuracy of most studies falls within the range of 90–99%, with some contamination detection and classification tasks achieving test set accuracy rates exceeding 99%. Despite the generally high detection accuracy, these high accuracy rates are primarily observed in test sets, and performance on external validation sets needs further improvement to address real-world application scenarios. Finally, regarding technological maturity, most of the current technologies remain at the laboratory research stage and lack practical industrial applications. The main obstacles that hinder the extensive industrial application of CNN models in the field of food safety, especially in the real-time production environment, are mainly reflected in the following three aspects: The first is the lack of quality and quantity of data. The training of CNN models depends on a large amount of labeling data, and data collection in food safety inspection is usually difficult. With a wide variety of food products and complex testing standards, it is difficult to obtain samples of sufficient quality. The second is the limitation of real-time and computing resources. In the food production process, the inspection system needs to process high-throughput data and give results in real time. CNN models, especially those with deep network structures, are computationally complex and usually require powerful hardware support. Moreover, real-time processing requires systems with extremely low latency, which is often difficult to achieve in some low-cost industrial equipment. Finally, environmental change and uncertainty. There are usually dynamic changes and uncertain factors in the food production environment, such as lighting changes, background noise and object occlusion, etc., which may affect the image quality. It is difficult for CNN models to adapt to these changes in the production environment, thus leading to the occurrence of false detection or missing detection.

Despite the challenges mentioned earlier in the application of CNN technology in food safety detection, it still demonstrates numerous advantages in practical applications. First, compared to traditional food safety detection methods, CNN technology offers significant advantages in terms of efficiency and adaptability [20]. Traditional methods typically rely on complex sample preparation and manual operations (such as sample separation in chromatography), whereas CNN can perform end-to-end detection directly based on image data, allowing for rapid analysis without the need to damage food samples [80]. Traditional methods often depend on fixed rules or assumptions (e.g., linear correlation), which can be limiting when faced with the complexity of food types and contaminants. CNNs, on the other hand, can flexibly adjust network architectures and parameters for different tasks, demonstrating strong adaptability to various tasks. Moreover, CNNs can automatically extract multi-level features from data, ranging from low-level features such as edges and textures to higher-level features such as shapes and patterns, addressing the limitations of traditional methods in processing complex visual features [81].

In addition, we further note that the combination of hyperspectral imaging (HSI) and CNNs shows significant advantages in food safety inspection tasks [82]. First of all, HSI technology can not only capture the surface image of food, but also record its internal spectral characteristics, achieving a comprehensive analysis from the two dimensions of image and spectrum. This integration of multi-modal data makes the HSI-CNN model particularly good for complex tasks, such as detecting multiple contaminants or components simultaneously. By capturing subtle spectral changes in food, HSI is able to identify trace contaminants or residues, while CNN is good at processing these high-dimensional, complex multi-modal data, and automatically extracting features and classifying them, thereby improving the accuracy and reliability of detection [83]. However, traditional RGB images can only capture information in the visible range, mainly reflecting the surface color and texture of the food, and lack in-depth analysis of the internal composition. This makes RGB images limited in the detection of trace pollutants or complex components, which can easily be missed or misjudged. In contrast, HSI is able to collect data across a wider range of the electromagnetic spectrum, covering multiple bands from visible light to near infrared. This means that HSI can not only record the appearance characteristics of food, but also reveal its internal chemical composition and physical structure, providing more comprehensive data support for food safety testing [84]. In addition, although the multi-dimensional dataset generated by HSI is complex, through the powerful processing power of CNN, key features can be automatically learned and extracted, greatly simplifying the data analysis process. As a result, the HSI-CNN model shows greater flexibility and robustness in detecting accuracy and handling complex tasks [85].

### 3.2. The Integration of CNNs and ViT Applications

In recent years, the integration of CNN and Vision Transformer (ViT) has demonstrated strong potential in food safety detection, particularly in complementing feature extraction and global information capture capabilities [86]. ViT uses self-attention mechanisms to capture global features in food images, which is especially useful in complex food scenarios (e.g., mixed food types) where it can model the relationships between different regions more effectively. Combining CNN and ViT in food safety detection allows CNN to provide high-quality local feature extraction, while ViT contributes to modeling global relationships, resulting in a more complete representation of food image features [87]. Monisha et al. [88] proposed a hybrid model combining ViT and CNN. This hybrid model leverages ViT to capture the global features of images, and then CNN refines these features, identifying local patterns of diseases to improve the accuracy of plant disease detection. Yang et al. [89] introduced an improved MobileViT (iM-ViT), a lightweight deep learning network that combines CNN and Transformer for the early detection of apple bruising. Compared to standard MobileViT, iM-ViT reduces the number of model parameters by introducing depthwise separable convolutions while merging both local and global features, enhancing the accuracy and robustness of bruising detection. Qi et al. [90] proposed a CNN–Transformer model combined with hyperspectral imaging technology to analyze the soluble solid content (SSC) and pH value of cherry tomatoes. Their research showed that integrating CNN with ViT enabled the collaborative use of global and local features, improving the accuracy of predicting SSC and pH values in tomatoes. Fu et al. [91] proposed a crop pest and disease image recognition algorithm based on an improved ViT method, aimed at enhancing the speed and accuracy of pest and disease detection in crops. The paper mentioned that future research may consider integrating CNN into ViT-based pest and disease recognition models, utilizing CNN’s inductive bias to reduce the amount of training data required, thereby lowering model training costs.

To sum up, the combination of CNN and ViT shows obvious advantages and improves detection accuracy. By integrating the local feature extraction capabilities of CNN and the global information modeling capabilities of ViT, this hybrid model can capture the details in food images and the relationships between different regions, thus effectively addressing the challenges posed by food images with diverse compositions, complex backgrounds, and possibly overlapping or obscuring objects [92]. When processing complex images, CNNs first function as front-end modules, focusing on extracting low-level features of the image, such as edges and textures. With its multi-layered structure, CNNs are able to abstract higher-level shapes and patterns layer by layer, helping to filter out irrelevant background interference. Next, the feature map generated by the CNN is segmented into patches of fixed size and converted into a one-dimensional sequence input into the ViT. ViT can process these image patches the same way it processes natural language, learning how they relate to each other, i.e., building a global context understanding of the image [93]. Although the research on the application of ViT and CNN in food safety detection is still in its infancy, some studies have shown that this fusion strategy can effectively make up for the limitations of a single model and provide a new solution for complex food image analysis.

Although the combination of CNN and ViT brings many advantages, there are some limitations to this combination approach. First, merging the two architectures may increase the complexity of the model, resulting in higher computational costs and longer training times. Second, while the introduction of ViT helps capture global features, in some cases, it may not be sufficiently sensitive to local features, especially when finely differentiated similar food types or disease situations are required [94]. As Fu et al. [91] point out, although the amount of training data required can be reduced by utilizing the inductive bias capability of CNNs, larger labeled datasets are still needed to adequately train deep learning models in order to achieve optimal performance. Finally, for real-time applications, the deployment of such hybrid models can be affected by hardware resource constraints, especially when running on mobile devices or embedded systems, where model size and inference speed are important factors that must be considered. Therefore, in practical applications, researchers and engineers need to weigh model performance against resource consumption to ensure that the solution is both efficient and practical.

In the future, with the continuous progress of hardware and algorithms, the CNN-ViT hybrid model will be further optimized to significantly improve the efficiency of real-time food safety inspection. In terms of hardware, GPU, TPU, and FPGA [95] are used to accelerate the training and reasoning speed of the model and enhance the real-time processing capability. In terms of algorithm optimization, pruning, quantization, and knowledge distillation techniques are used to reduce model parameters and computational complexity, so as to improve operational efficiency. In addition, the design of lightweight and efficient network structures, such as combining lightweight CNN modules with simplified versions of ViT, while maintaining high accuracy and reducing computational requirements, will further promote the wide application of the model [96]. These hardware and algorithm improvements will not only improve the performance of the CNN-ViT hybrid model, but also make it more suitable for resource-constrained environments, ensuring efficient and accurate food safety detection on edge devices.

## 4. Food Safety Risk Prediction Using RNNs

This section primarily discusses the application and characteristics of traditional recurrent neural networks (RNNs) and their variant Long Short-Term Memory (LSTM) networks in food safety risk prediction. First, by elaborating on the time series modeling advantages of RNNs, it analyzes their specific performance in food safety risk prediction and highlights the limitations of RNNs due to the issues of gradient vanishing and explosion. Next, it explains how LSTM addresses the shortcomings of RNNs by introducing memory units and gating mechanisms, and demonstrates its widespread application and technical advantages in various areas such as environmental condition forecasting, risk warning, and identification. Furthermore, this section also conducts a comparative analysis of LSTM’s performance in practical applications, pointing out its limitations in terms of computational cost and interpretability, and suggesting directions for further optimization.

### 4.1. Conventional RNNs

Food safety risk prediction plays a critical role in food safety. As global supply chains become increasingly complex and food production, processing, and transportation become more diverse, food safety issues have become more hidden and complex [97]. Early prediction of potential risks can help regulatory authorities, businesses, and consumers take effective preventive measures before food safety issues occur, reducing the likelihood and impact of incidents [98]. Food safety risk prediction can identify risk factors in the production process and assess the potential threats posed by external environmental changes (e.g., temperature and humidity fluctuations, consumer sentiment changes) to food safety, offering a more comprehensive risk assessment [99]. Additionally, food safety risk prediction can optimize resource allocation and enhance the efficiency of food safety management. By analyzing historical data and real-time monitoring data, predictive systems can identify possible sources of risk, thereby helping businesses adjust production processes, strengthen quality control, and avoid potential food safety incidents [100].

RNNs are particularly suitable for food safety risk prediction because of their cyclic structure, which can effectively handle time series data and sequence-dependent tasks [101]. The core of RNNs is their recursive feedback mechanism, which enables the networks to not only receive the input of the current time step, but also remember the output of the previous time step, thus capturing information about dynamic changes and long-term dependencies in the data [43]. This enables RNNs to predict and identify potential food safety risks in advance when monitoring key parameters such as the operating status of production equipment, temperature and humidity in the storage environment, and temperature in cold chain logistics in real time [43]. There are key differences between RNNs and traditional approaches when managing multivariate time series data for food safety risk prediction. RNNs are able to remember historical information and make predictions in conjunction with current inputs, automatically learning nonlinear dynamic relationships, and are particularly suitable for dealing with complex factors in the food supply chain (e.g., temperature, humidity, transportation conditions) without the need for strict mathematical assumptions. At the same time, RNNs support interaction modeling among multiple variables, reduce the dependence on artificial feature engineering, and can realize real-time prediction and adaptive updates. In contrast, traditional statistical modeling methods (such as ARIMA and VAR) rely on stationarity and linear relationship assumptions, which makes it difficult for them to deal with the nonlinear interaction effects common in the food supply chain [102]. Although traditional machine learning methods (such as SVMs and decision trees) can deal with nonlinear relationships, they lack the modeling ability of order dependence, which easily leads to information loss or overfitting [103]. In addition, they often need to be retrained to deal with new data, and the response is lagging, which makes it difficult for them to meet the dynamic and real-time requirements of food safety risk prediction.

RNNs’ ability to process time series data allows for the dynamic modeling of individual processes in the food supply chain, particularly where historical data and contextual information are considered. Whether conducting an early warning of food safety hazards or performing a risk assessment, RNNs provide effective forecasting and analysis. Zhong et al. [104] proposed an integrated algorithm framework based on AHP-EW and AE-RNN for the early warning and analysis of food safety hazards. In their study, the RNN model was used to process food inspection data with time series characteristics, receiving feature vectors extracted by an autoencoder and then performing nonlinear transformations to output risk prediction results. The paper also noted that compared to traditional statistical-based time series forecasting methods, the AE-RNN model can predict data more accurately and performs better when handling high-dimensional and complex data. Zhao et al. [105] introduced a risk assessment model for agricultural products based on an enhanced recurrent neural network (RNN). This model, based on RNNs, can capture both long-term dependencies and short-term fluctuations in agricultural product price data, providing a basis for future risk prediction. Harrag et al. [106] developed a deep learning-based event extraction system designed to detect and extract food hazard-related events from Arabic texts on social media. In their paper, RNNs were used to process and predict context-based outputs, which are critical for event prediction and extraction. Additionally, the RNN model can identify and classify entities within the text, such as locations, quantities, dates, etc., which are particularly important for risk warning systems.

By comprehensive comparison of the above research results, it can be found that in the current task of food safety risk prediction, all studies have made full use of the advantages of RNN in processing time-dependent and context-dependent data. With its memory ability and the ability to model continuous data, RNN provides an effective solution for food safety incident monitoring, risk quantification, and continuous prediction. However, RNNs also expose some technical shortcomings in the face of long time series data, especially gradient disappearance and gradient explosion problems, which make it difficult for the model to capture long-term dependencies. To solve this problem, Zhao et al. [105] introduced the adaptive computation time (ACT) mechanism, which effectively improved the performance of RNN in long-term dependent data modeling. Zhong et al. [104] further enhanced the feature learning and representation ability of RNN for high-dimensional complex data by combining the autoencoder (AE). These improvements provide a new direction to address the inherent shortcomings of RNNs. However, there are still some gaps in risk assessment and prediction. For example, most current models rely on a single data source and fail to effectively integrate heterogeneous data from multiple sources (such as text data, food detection data, sensor monitoring data), resulting in the limited application of models in cross-domain risk monitoring and early warning. In addition, the lack of a unified risk prediction framework makes it difficult to achieve comprehensive dynamic monitoring of food safety incidents. Future studies can combine the modeling advantages of RNN to propose a unified framework integrating data mining, risk prediction, and time series analysis, break through the barriers of multi-source data, and realize the cross-domain monitoring of food safety incidents and dynamic risk prediction, so as to provide a more intelligent and real-time solution for food safety supervision.

### 4.2. Variants of RNNs: LSTM

However, RNNs have significant limitations, such as the tendency to encounter vanishing or exploding gradient issues when predicting long time series, making it difficult for the model to effectively capture long-term dependencies and global information. Additionally, RNNs have high computational overhead, especially when processing long sequences of data, which may affect the efficiency of real-time predictions [44]. In food safety risk prediction, these limitations manifest in the inability to accurately understand the impact of long-term historical data on future risks. For example, when detecting potential hazards in the food processing chain, RNNs may struggle to recognize key events that occurred several steps earlier.

Long Short-Term Memory (LSTM) networks address these issues effectively by introducing forget gates, input gates, and output gates. The core of LSTM lies in using memory cells to store and update information over time, thereby alleviating gradient problems [107]. Specifically, LSTM is capable of extracting deep patterns from complex time series data, selectively forgetting irrelevant information while retaining important data, enabling it to better capture the interactions between long-term and short-term factors in food safety risk prediction. For example, when predicting risk changes during the shelf life of food, LSTM can more accurately analyze the combined effects of multi-step processes, storage conditions, and historical monitoring data [108]. Additionally, LSTM is more robust to noise and outliers, performing exceptionally well in handling nonlinear changes and multivariate inputs, and can model dynamic relationships between key factors at different stages (e.g., raw material supply, processing, transportation).

Traditional time series analysis methods can only model short-term dependence, and it is difficult for them to capture the long-term dynamic changes in the food supply chain. In addition, as we discussed earlier, traditional methods are limited in their ability to deal with high-dimensional data and often assume linear relationships between variables, making it difficult for them to make effective predictions in the face of complex nonlinear relationships between variables [109]. In contrast, LSTM-based models, with their unique gating mechanisms and real-time prediction and dynamic update capabilities, can effectively capture long-term dependencies in time series and automatically learn nonlinear dynamic patterns. In terms of multivariate modeling, LSTM can not only process multiple time series variables simultaneously, but also capture complex interactions between variables, such as the combined effects of temperature, humidity, and transportation time. Therefore, the application of LSTM significantly improves the accuracy and reliability of food safety risk prediction, providing stronger decision-support capabilities for enterprises and regulatory bodies [110]. Table 7, Table 8, Table 9 and Table 10 summarize the current state of LSTM applications in food safety risk prediction, covering four main directions: public opinion analysis, supply chain demand forecasting, environmental condition prediction, and risk warning and analysis. The content is detailed and compared across multiple dimensions, including target tasks, technical methods, dataset sources, prediction accuracy, and technological maturity. Overall, the tables highlight the broad application potential of LSTM in time series forecasting and its advantages in food safety risk prediction.

As shown in the tables, LSTM has demonstrated its powerful time series modeling capabilities and broad applicability in food safety risk prediction. From the method dimension, LSTM is often applied not only as a standalone model but also in combination with other models (such as BERT, GBR, etc.) to enhance performance. In terms of datasets, the data sources across various directions are diverse, including social media, industry-standardized data, and sensor data, each with different structural features and preprocessing requirements. In terms of accuracy, LSTM’s ability to handle multivariate time series data allows it to perform exceptionally well across multiple tasks, such as achieving an F1 score above 97% in public opinion analysis and reducing the average error in supply chain forecasting to as low as 6.56%. Regarding technological maturity, some directions, such as public opinion analysis and supply chain forecasting, have already entered the practical application stage, while others remain in the laboratory research phase.

The advantage of LSTM lies in its ability to capture long-term dependencies, making it suitable for modeling dynamic and complex systems, while exhibiting good robustness on both structured and unstructured data [126]. However, its limitations should not be ignored. On the one hand, LSTM may overfit when dealing with small datasets, affecting the generalization ability of the model, and its “black box” nature also limits the interpretability of prediction results, weakening its trust in high-risk decision-making scenarios. On the other hand, LSTM faces many challenges in resource-constrained environments. First, LSTM requires high computing and storage resources when processing long time series or high-dimensional data, which can lead to inefficient operation or even failure to deploy. Secondly, although LSTM alleviates the gradient disappearance problem to a certain extent, its parameter optimization ability is limited in the case of limited resources, and it is still difficult to capture remote dependencies [127]. In addition, LSTM needs to store a large number of intermediate states and weight parameters, and in multivariate time series analysis, memory requirements can increase exponentially, posing a serious challenge to devices with limited storage capacity. Therefore, combining LSTM with attention mechanisms or graph neural networks, and optimizing its structure for domain-specific characteristics, may be a promising direction for improving its performance in the future [128]. The attention mechanism can focus on the important information in the time series, reduce the processing of redundant data, and directly calculate the correlation between different time steps, which solves the limitation of LSTM in capturing remote dependencies in long time series [129]. GNNs can efficiently model complex nonlinear relationships among multiple variables through graph structure, avoiding the high computational overhead of LSTM in capturing variable interactions. At the same time, GNNs support distributed computing, which can reduce the computing burden of a single device through local graph operations and adapt to low-resource environments [130]. Combined with the attention mechanism of the graph neural network, they can also dynamically identify the most important relationships between variables, focus computing resources on key nodes and edges, and further improve efficiency. Overall, LSTM provides a precise and reliable tool for food safety risk prediction but still requires further optimization to address challenges in real-world applications.

In the future, in order to improve the computational efficiency and scalability of LSTM model in practical applications, the development of hardware cannot be ignored. By optimizing the memory architecture, such as expanding on-chip memory capacity and increasing memory bandwidth, data transmission latency can be effectively reduced. Using low precision calculation can not only reduce the computational complexity of LSTM model, but also maintain the model’s performance. In addition, combined with the cooperative work of CPU, GPU, and NPU (neural processing unit), computing resources can be dynamically allocated according to task requirements, making full use of the advantages of different hardware, thus enhancing the parallel processing capability of the system. Together, these improvements are designed to ensure that the LSTM model achieves greater efficiency and scalability while maintaining high performance.

## 5. Future Outlook and Challenges

This section discusses the future development and challenges of deep learning models, such as CNN and RNN, in the field of food safety. It explores the integration of these models with technologies like the Internet of Things (IoT), blockchain, and federated learning, detailing the methods of integration, potential challenges during the integration process, and corresponding solutions. Additionally, it analyzes the key challenges regarding model interpretability, including the complexity of the decision-making process and insufficient adaptability to industry regulations. This section proposes solutions such as adopting Explainable Artificial Intelligence (XAI) technologies and integrating models with logical rules to enhance transparency and practical utility. Figure 6 illustrates the challenges, limitations, and future development directions of deep learning technologies like CNNs and RNNs in applications.

### 5.1. Integration with Other Technologies

#### 5.1.1. Internet of Things (IoT) Technology

IoT (Internet of Things) technology connects sensor devices with food safety management systems to enable real-time data collection, transmission, and analysis throughout the food supply chain. IoT provides the multi-modal data inputs required for deep learning models, and when combined with CNN’s feature recognition capabilities and RNN’s time series prediction advantages, it enables real-time monitoring of issues in food production, processing, and transportation, as well as accurate prediction of potential risks [131].

IoT devices are equipped with sensors to collect multi-modal data related to the food supply chain in real time, such as temperature and humidity, gas composition, vibration, and spectral information, providing a continuous data flow for subsequent analysis. When IoT technology is integrated into the model, CNNs can make full use of image data collected by sensors (such as food appearance, microbial detection images, or spectral information) to achieve real-time monitoring and rapid detection of problems such as corruption, contamination, or quality anomalies. RNNs, on the other hand, focus on processing time series data generated by sensors (such as temperature and humidity changes, vibrations during transportation, and environmental parameter fluctuations) to capture dynamic trends and potential anomalies in real time and predict food safety risks under different environmental conditions [132]. In this integration process, the IoT network enables the efficient transfer of sensor data to edge computing devices or cloud platforms, where CNNs and RNNs can process and analyze the data in depth. This seamless data collection and intelligent analysis process helps improve the accuracy and efficiency of food safety inspection and risk prediction. In addition, the integration of CNNs and RNNs with IoT can also be used for food supply chain traceability and optimization. By recording data from the entire process of food production, processing, transportation, and sales through IoT devices, CNNs can quickly identify food production batches, quality levels, or package integrity, while RNNs can analyze trends in the logistics environment, predict possible quality issues, and optimize transportation routes. Edge computing supported by IoT technology further improves detection efficiency [133]. With this integrated system, the inspection process is more real-time and preventive, while covering the whole chain management from production to consumption. This combination of technologies improves the level of intelligence and automation of food safety.

However, the integration of deep learning models with IoT also presents certain challenges. The collection and management of data are core issues for integrating Internet of Things (IoT) sensors in the food supply chain. With the widespread deployment of sensors in all aspects of production, processing, and transportation, the explosion in data volume increases the complexity of collection, storage, and management. Large-scale food supply chain systems require the deployment of a large number of sensors, and the continuous operation of cloud servers also requires the processing and storage of high-dimensional image data and long time series data, which not only increases hardware costs, but also brings data management and processing challenges. In addition, the effectiveness of IoT technology is also closely related to the distribution density of sensors. In food safety testing, from production lines to cold chain transportation to storage environments, different scenarios have different needs for the number and layout of sensors [134]. If the sensor distribution is too sparse, it may result in insufficient data coverage to fully monitor food quality. If it is too dense, it will further increase the complexity of data management and processing. Therefore, in the combination of IoT and deep learning, it is particularly important to balance the comprehensiveness of data collection with the feasibility of processing while optimizing the deployment strategy of sensors.

#### 5.1.2. Blockchain Technology

The core features of blockchain technology, such as decentralization, immutability, and transparency, ensure the authenticity and integrity of data across all stages of the food supply chain. Deep learning models, on the other hand, process and analyze large volumes of data to extract valuable patterns, thereby enhancing the reliability of food safety detection and risk prediction results [135].

By recording every stage from production to consumption on the blockchain, deep learning models can analyze these data to optimize logistics routes, inventory management, and demand forecasting within the supply chain, thereby improving the overall efficiency of the supply chain [136]. This is particularly crucial in the event of a food safety incident, as deep learning can quickly analyze the historical data accumulated on the blockchain to identify contaminated or non-compliant product batches. Furthermore, key data from each link in the supply chain can be recorded on the blockchain, ensuring that the data remain immutable and transparent [137]. For example, in the food production or transportation link, detection data can be recorded in real time through the blockchain platform and time-stamped to ensure the authenticity and traceability of the data. The distributed ledger of blockchain integrates data from different links (such as production, transportation, retail), ensuring that information is shared and immutable, providing a reliable source of data for CNNs and RNNs, thereby improving the accuracy of model training and inference. With smart contracts, RNN-based predictions can trigger alerts or decisions in real time, such as freezing problematic batches or notifying interested parties to take action. Combined with the traceability ability of blockchain, when CNNs detect a problem with a batch of food, they can quickly trace back to the specific production or transportation link, quickly locate the root cause of the problem, and reduce the scope of food safety accidents. This integrated approach provides food companies and regulators with a transparent and efficient solution to ensure food safety throughout the supply chain.

However, deep learning models require large amounts of high-quality data for training, and directly storing all data on the blockchain would increase the storage burden of the blockchain. Additionally, the consensus mechanisms of blockchain (such as Proof of Work (PoW) or Proof of Stake (PoS)) limit the speed at which transactions can be written [138], and in food safety scenarios where real-time decision-making is critical, data delays could prevent deep learning models from providing timely predictions. Therefore, one potential solution could be to store only summarized data (e.g., model prediction results) on the blockchain, while storing raw data (such as sensor data or training datasets) in off-chain databases [139]. For example, in cold chain logistics, the complete time series data of temperature and humidity could be stored in off-chain databases, while the marked key anomalies and deep learning model predictions could be stored on the blockchain [140].

#### 5.1.3. Federated Learning

Federated learning, through a distributed training approach, allows for the collaboration of data from various nodes within the food supply chain. This can further expand the application of deep learning models, enabling distributed model training while ensuring data privacy and security [141]. This approach is suitable for multi-node collaborative modeling in the food supply chain, enhancing the accuracy and robustness of food safety detection and risk prediction.

Integrating federal learning with CNNs and RNNs into food safety applications also presents multiple challenges. Data heterogeneity is a core issue, as data in the food supply chain may come from different companies, links, or geographic regions. For example, some companies have different food inspection equipment or data collection methods, causing CNNs and RNNs to face distribution differences when processing these data. In addition, federated learning requires repeated transmission of model parameters between participants, while nodes in the food supply chain may be widely distributed, with limitations in network bandwidth and communication latency [142]. CNNs and RNNs usually have large models and a large number of parameters involved, which makes communication costs significantly increase. Finally, under the federated learning framework, nodes may differ in computing power, data volume, and availability. For example, nodes in a food production enterprise may have more detection data, while nodes in some small transportation enterprises have less data. This imbalance of participants will lead to the unstable training of the federated model and affect overall performance.

In federated learning, deep learning models can be locally trained at different nodes in the food supply chain without directly sharing data. Instead, the model parameters are exchanged, allowing for federated learning [143,144]. Different nodes in the supply chain use their respective data to train the distributed deep learning models. This process can generate various types of data, such as image data from the production process and time series data during transportation, and federated learning can leverage CNNs and RNNs to simultaneously process multi-modal data. Additionally, blockchain can serve as a trusted platform to coordinate this process, ensuring data privacy while enhancing the model’s predictive and decision-making capabilities [145].

### 5.2. Limitations of CNNs and RNNs

In the field of food safety, the application of CNNs and RNNs faces several limitations and challenges. From the table content analysis and comparison in Chapters 3 and 4, it can be seen that the limited size and diversity of datasets directly affect the generalization ability of CNNs and RNNs in the field of food safety. First, insufficient data size can easily lead to overfitting of models, and CNNs and RNNs may remember specific features in the training data and fail to generalize patterns that apply to the new data. For example, CNNs may only learn the characteristics of food contamination under certain light conditions and perform poorly in the face of different light conditions or types of contamination. Similarly, RNNs may rely too heavily on specific transport patterns for time series data, making accurate predictions only for common patterns in the training set, but not for new environments or anomalies. At the same time, limited datasets tend to lead to data distribution bias, and certain categories in the training set may be over-represented, while other important but scarce categories are ignored. In this case, CNNs may be focusing too much on frequent pollution features while ignoring rarer but critical safety hazard features. RNNs may only perform well for the common time series patterns in the training set and have poor prediction accuracy for the less common abnormal patterns. This distribution bias reduces the generalization ability of the model, especially when dealing with rare high-risk scenarios in food safety, where the model performance is often insufficient.

In addition, in the field of food safety, the limited scope of the model is also a major challenge. Food safety scenarios often involve multi-modal data, including images, time series, sensor inputs, and text records [50]. The diversity of data makes it difficult for a single type of deep learning model to fully meet all requirements. CNN has low efficiency in processing time series or other non-image data, and it is difficult for it to deal with the complex food supply chain environment alone, especially in the multi-modal data fusion scenario, where its performance is limited. RNNs and their variants LSTMs also face adaptation problems. For example, dynamic data on food safety often change significantly depending on region, season, or food type, and RNNs are sensitive to these changes in distribution and often need to adjust or retrain the model frequently to adapt to new scenarios. These limitations make it difficult for these two models to meet the diverse needs of complex food safety applications alone.

The high dependence of CNNs and RNNs on high-quality training data is another limitation in the field of food safety. The effectiveness of CNNs relies on large and clearly labeled image datasets [146], but high-quality food contamination images or abnormal packaging data in the food industry are often costly and difficult to obtain and label. In addition, timing data (such as temperature and humidity) in the food supply chain may be missing or inaccurate due to differences in data acquisition equipment, and RNNs will often experience performance degradation or even prediction errors when processing these low-quality or incomplete data. For these two models, the lack of high-quality labeled data directly leads to inadequate training, which affects the prediction accuracy and practicability of the models.

### 5.3. Interpretability of the Model

Deep learning models, such as CNNs and RNNs, are often applied in food safety scenarios directly related to critical decision-making, such as automating the detection of food surface defects with CNNs or predicting risks in cold chain logistics with RNNs. However, the decision-making process of deep learning models is complex and highly nonlinear, making it difficult to explain how specific features influence the prediction results [147]. In the field of food safety, many food safety events (such as expiration, deterioration, etc.) are highly uncertain and low-frequency. These high-risk events are often rare in the training data, and as a result, it is difficult for models to provide clear explanations when these events occur. Especially in the case of predictive errors in the model, it is difficult to trace back to the specific cause, especially when the food safety problem involves multiple factors (such as packaging defects, transportation conditions, etc.); how the model integrates this information to make decisions is often a black box process. This “black box” nature of deep learning models makes their decision processes hard to interpret, which not only limits their trustworthiness in high-risk scenarios but also poses challenges for regulation and accountability [148]. In cold chain logistics or food production lines, incorrect decisions can lead to widespread food contamination or quality issues. If the model cannot explain the source of its results, it becomes challenging for users to trust the predictions and take appropriate actions [149].

Furthermore, the food safety industry is governed by strict regulations and standards [150] (e.g., temperature and humidity ranges for cold chain logistics, requirements for food packaging integrity). However, the prediction results of deep learning models often lack clear connections and explanations relative to these rules, making it difficult for their outcomes to be directly applied in real-world scenarios. In the food supply chain, businesses generally need to take specific actions based on industry rules, and if deep learning models’ predictions cannot be directly translated into actionable recommendations, their practical utility is compromised [151]. In food safety regulation, if the model’s predictions cannot explain the specific issues, it may lead to disputes in accountability and traceability, thereby reducing the model’s trustworthiness. First, a precise demarcation of responsibilities is essential to ensure that stakeholders fulfill their statutory responsibilities. When using CNNs and RNNs for food safety testing and risk prediction, the lack of model interpretability significantly increases the difficulty in defining responsibility. Specifically, when a food safety incident occurs, it is difficult for regulators to clearly identify through these models whether the source of the problem is the raw material, the production process, or the logistics, which not only impedes effective accountability mechanisms, but can also lead to legal disputes and undermine trust in the outcome of decisions across the supply chain. Second, the food safety traceability process requires that the source of contaminated food be quickly and accurately traced so that corrective action can be taken in a timely manner. However, the predictions of deep learning models are based on complex pattern recognition, which makes it difficult for them to provide intuitive causal explanations. Even when models can accurately predict risks, regulators have difficulty understanding the basis for their decisions, which can result in missing key information points and slow response, leading to wrong management decisions. Therefore, while CNNs and RNNs excel in image and sequence data analysis, their “black box” nature limits their wide application. Models cannot clearly explain their predictive logic, and even with high accuracy, may be questioned for lack of transparency.

To address the issue of interpretability in deep learning models, Explainable Artificial Intelligence (XAI) techniques can be employed as a supplementary tool. The core goal of XAI technology is to reveal the internal decision-making process of deep learning models, thereby making their application in food safety more transparent and trustworthy [152]. By visualizing and analyzing the feature extraction and prediction mechanisms of models such as CNNs and RNNs, XAI techniques help users understand how the model arrives at specific decisions [153]. One of the XAI techniques, Grad-CAM, generates heatmaps that highlight the most important areas in an input image that contribute to the deep learning model’s (e.g., CNN) specific prediction, enhancing the model’s interpretability [154]. For instance, in the task of food surface defect detection, Grad-CAM can visually show how CNNs determine a food item to be defective, such as highlighting areas with mold spots or surface cracks. For time series data processed by RNNs, another XAI technique, Saliency Maps, can be used to mark the importance of features at key time points. For example, in temperature and humidity prediction for cold chain logistics, Saliency Maps can highlight which time periods’ abnormal changes led the model to predict spoilage risk [155]. Additionally, attention mechanisms can dynamically allocate computational resources in the model, focusing on key parts of the input data. In RNNs, attention mechanisms can explicitly show which time steps contribute most to the final prediction [156]. For instance, when analyzing cold chain logistics data, the attention mechanism might emphasize certain periods of temperature and humidity anomalies, helping to explain how the model determines food spoilage. Mahmud et al. [157] proposed a deep transfer learning-based tomato leaf disease detection method, where they used Grad-CAM to visualize the regions of the image that the CNN model focused on. Grad-CAM calculates the gradients of convolutional layer feature maps and weighs these gradients with activations for a specific class to generate a heatmap, showing the areas in the input image most influential to the classification result. Their study demonstrated that Grad-CAM enhanced the interpretability and transparency of the CNN model in the tomato leaf disease detection task. Karim et al. [158] applied Grad-CAM for real-time grape leaf disease classification, identifying and highlighting the affected areas of grape leaves. Grad-CAM can also be used in real-time operations to precisely locate areas requiring specific treatment measures, such as selective pruning or targeted pesticide application. This assists farmers and agricultural experts in precisely identifying and addressing the areas that need attention, thus increasing trust in the model’s predictions. Jin et al. [159] utilized near-infrared hyperspectral imaging combined with deep learning methods (LeNet, GoogLeNet, and ResNet) to identify different rice seed varieties. They used the Saliency Maps method to visualize CNN models and enhance model interpretability. Saliency Maps help in understanding the decision-making process of the model by highlighting the regions of input data that most influence the model’s prediction. Adak et al. [160] compared simple and hybrid deep learning techniques (LSTM, Bi-LSTM, Bi-GRU-LSTM-CNN) for sentiment analysis and used XAI techniques—SHAP and LIME—to explain model predictions. SHAP predicts instances by calculating the contribution of each feature to the prediction, while LIME focuses on training local surrogate models to explain individual predictions. Their research showed that combining LIME and SHAP techniques with the LSTM model not only enabled high-accuracy sentiment analysis but also provided interpretability for the model’s predictions.

For the acceptance of deep learning models by consumers and practitioners, two key strategies can be employed. On the one hand, continuously optimizing models and refining their application domains can improve the accuracy of deep learning applications in food safety, thereby preventing trust crises among consumers and lowering the technical barriers for practitioners using these technologies. On the other hand, educating both consumers and practitioners about relevant knowledge can gradually increase consumer trust and help practitioners gain the necessary skills. This, in turn, can alleviate concerns about artificial intelligence replacing certain job functions, thereby enhancing the overall acceptability of deep learning technologies within the industry.

## 6. Conclusions

This review provides a critical analysis and systematic overview of the applications of convolutional neural networks (CNNs) and recurrent neural networks (RNNs) in food safety detection and risk prediction. This paper summarizes the research progress of these two deep learning technologies in the field of food safety and discusses their technical advantages and limitations in various application scenarios. The literature sources are extensive, covering multiple domains such as food safety detection, risk prediction, image recognition, and time series analysis, demonstrating the widespread application and profound impact of deep learning technologies in the food safety sector.

In the area of food safety detection, CNNs are primarily used for processing and feature extraction from image data, showing significant advantages in scenarios like food appearance inspection, surface defect identification, and adulteration analysis. RNNs and their variant, LSTM, excel in handling time series data, making them suitable for predictive time series modeling, such as the prediction of contamination risks in production environments and real-time monitoring of food spoilage in cold chain logistics. Through comparative analysis, this review reveals the technical characteristics and performance of CNNs and RNNs in different application areas, offering references for future research. The literature mentions that integrating deep learning technologies with traditional methods (e.g., combining CNNs with hyperspectral imaging and RNNs with statistical time series prediction methods) has proven to significantly enhance food safety detection and risk prediction outcomes. Additionally, the integration of deep learning models with machine vision technologies can automate quality control processes on food production lines, improving detection speed and accuracy.

Current research focuses on improving detection accuracy, optimizing model structures, and enhancing the real-time performance and adaptability to various data types. Future challenges include how to more effectively integrate these deep learning models with other technologies (such as IoT, blockchain, and federated learning) to ensure more comprehensive food safety management. Model interpretability is also a key challenge, and this paper suggests that Explainable Artificial Intelligence (XAI) techniques can be used to enhance the transparency and trustworthiness of models.

In conclusion, CNNs and RNNs have broad application prospects in the field of food safety. This is mainly due to the huge amount of data faced by the contemporary food industry and the urgent need for high standards of automation. With the wide application of smart sensors, Internet of Things devices, and data acquisition technology, the food industry has accumulated massive data in production, transportation, storage, and other links. It has been difficult for traditional manual detection and manual analysis methods to cope with these complex data processing needs, and at the same time, real-time monitoring and timely warning are also crucial for preventing food safety accidents. In this context, deep learning models such as CNNs and RNNs show significant advantages. On the one hand, they can effectively handle large-scale, complex types of data. On the other hand, they support automated real-time detection and prediction, which can well meet the high requirements of food safety in the food industry. Compared with traditional technologies, these models can not only significantly improve the accuracy of food safety detection and risk prediction, but also greatly improve the detection efficiency. Therefore, the application of CNNs and RNNs in the field of food safety has become an indispensable and important technical means. Through continuous technological innovation and optimization, these deep learning models are expected to play a greater role in food safety management in the future, providing strong technical support for safeguarding public health and social stability.

## Figures and Tables

**Figure 1 foods-14-00247-f001:**
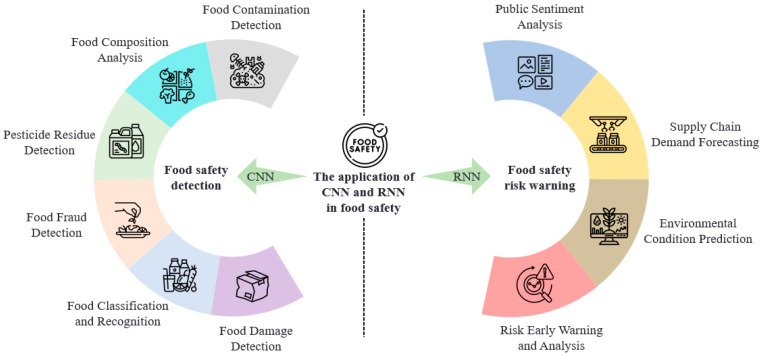
Application of CNNs and RNNs in food safety.

**Figure 2 foods-14-00247-f002:**
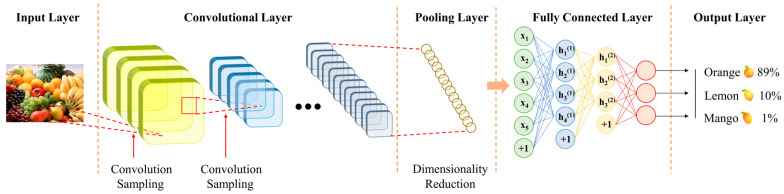
Basic architecture of CNNs. “•••” denotes omitted convolutional layers, arrows indicate data flow and processing steps.

**Figure 3 foods-14-00247-f003:**
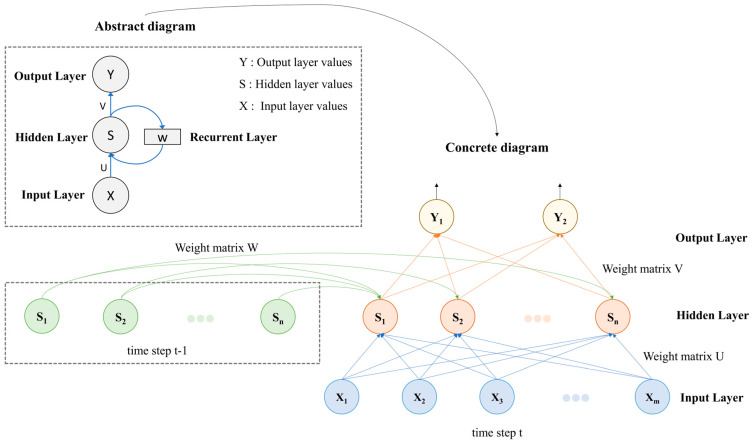
Basic architecture of an RNN. Arrows indicate data flow and processing steps, subscripts denote different nodes, different colors of “•••” indicate that intermediate nodes are omitted at different time steps or in different layers.

**Figure 4 foods-14-00247-f004:**
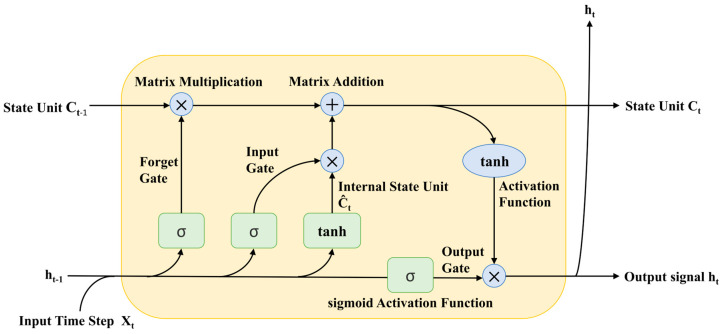
Internal structure of an LSTM.

**Figure 5 foods-14-00247-f005:**
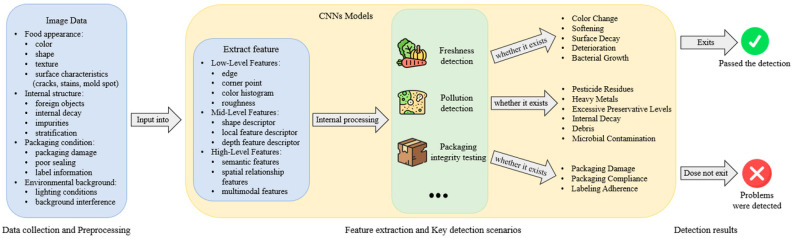
Workflow of food safety testing using CNN. “•••” indicate more application scenarios for food safety testing.

**Figure 6 foods-14-00247-f006:**
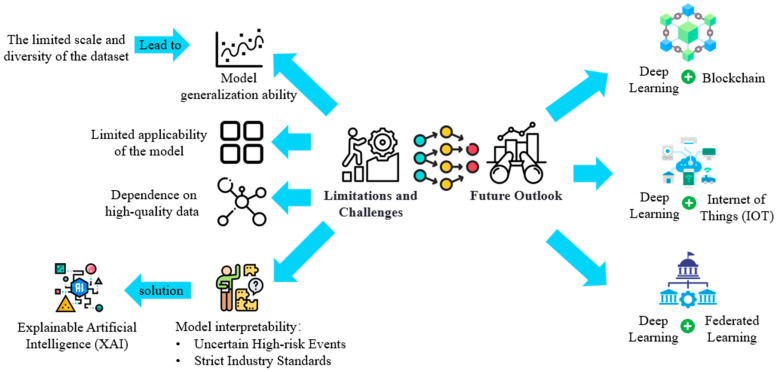
Future outlook, limitations, and challenges.

**Table 1 foods-14-00247-t001:** Applications of CNN in food contamination detection.

References	Objective	Analyses	Equipment	Dataset	Accuracy	Technology Maturity
[53]	Detection of the Contamination Status and Types of Foodborne Pathogens in Mutton	SWIR-HIS, 1D-CNN, PLS-DA, SVM	SWIR-HSI Imaging System	210 Uncontaminated Sample Images and 210 Contaminated Sample Images with Different Concentrations	Test Set: 92.86%, External Validation Set: 97.62%	Laboratory Phase
[54]	Detection of Contamination in Food and Medical Packaging	DCNN	Electromechanical Platform (Equipped with Time-of-Flight Sensor)	2700 RGB Images of Nine Types of Contaminants (e.g., Coffee, Spots, Chocolate, etc.)	99.74%	Laboratory Phase
[55]	Investigation of the Relationship Between Fungal Contamination and Germ Orientation in Maize Kernels	Vis-NIR HIS, PLS-DA, 1D-CNN	Visible to Near-Infrared Hyperspectral Imaging System	Two Sets of 200 Maize Kernels (100 Sterile and 100 Contaminated) Captured in Different Orientations	Test Set: 99.36%, External Validation Set: 98.04%	Laboratory Phase
[56]	Real-Time Detection of Foreign Contamination in Food Packaging	HIS, CNN	Hyperspectral Camera	A Total of 210 Images Including 150 Contaminated and 60 Normal Sealed Trays	Above 94%	Laboratory Phase
[57]	Detection and Quantification of Clostridium Spore Contamination in Food	HIS, 1D-CNN, RF	HSI System Comprising a Hyperspectral Camera	4250 Images of Bacillus cereus Spores at Different Concentrations	90–94%	Laboratory Phase

**Table 2 foods-14-00247-t002:** Applications of CNN in food composition analysis.

References	Objective	Analyses	Equipment	Dataset	Accuracy	Technology Maturity
[58]	Detection of Dry Matter Content in Potatoes	Vis/NIR, 1D-CNN	Halogen Lamps, Spectrometer, PLC System	Spectral Images of 100 Potato Samples	R^2^P = 0.934	Laboratory Phase
[59]	Prediction of Fat Content in Salmon Filets	NIR-HIS, RACNN	HSI System, Spectrometer	Spectral Images of 100 Salmon Samples	R^2^P = 0.9033	Laboratory Phase
[60]	Prediction of Linoleic Acid Content in Red Meat	HIS, CNN-Bi-LSTM	Hyperspectral Imaging System	Spectral Images of 252 Red Meat Samples	R^2^P = 0.909	Laboratory Phase

**Table 3 foods-14-00247-t003:** Applications of CNN in pesticide residue detection.

References	Objective	Analyses	Equipment	Dataset	Accuracy	Technology Maturity
[61]	Non-Destructive Detection and Identification of Pesticide Residues on Garlic Chive Leaves	SWIR-HIS, 1D-CNN	Short-Wave Infrared Hyperspectral Imager	Spectral Images of 30 Garlic Chive Leaf Samples	Training Set: 98.5%, Test Set: 97.9%	Laboratory Phase
[62]	Detection of Different Levels of Pesticide Residues in Grapes	Vis/NIR-HIS, CNN, ResNet, LR	Vis/NIR and NIR Hyperspectral Imaging Systems	Spectral Images of Three Grape Varieties Treated with Four Different Concentrations of Pesticides.	Above 93%	Laboratory Phase
[63]	Non-Destructive Detection of Pesticide Residues on the Surface of Hami Melons	Vis/NIR, 1D-CNN	Vis/NIR Spectrometer	Spectral Images of 120 Hami Melon Samples	95.83%	Laboratory Phase
[64]	Detection of Multiple Pesticide Residues on Chinese Toon Leaves	THz, DCNN	Terahertz Imaging System	Images of Nine Types of Chinese Toon Leaf Samples	Training Set: 97.27%, Test Set: 96.74%	Laboratory Phase
[65]	Rapid On-Site Detection of Pesticide Residues in Tea Leaves	SERS, 1D-CNN	Handheld Raman Spectrometer	Spectral Images of Four Tea Varieties Treated with Pesticide Solutions at Different Concentrations	Higher Accuracy Compared to Traditional Methods	Practical Application Phase

**Table 4 foods-14-00247-t004:** Applications of CNN in food fraud detection.

References	Objective	Analyses	Equipment	Dataset	Accuracy	Technology Maturity
[66]	Quantitative Detection of Adulterated Coffee Samples	CNN, FT-NIR	Fourier Transform Near-Infrared Spectrometer	Spectral Images of 550 Commercial Espresso Samples Mixed with Chicory, Barley, and Corn	R^2^ > 0.98	Laboratory Phase
[67]	Detection of Adulteration in Turmeric Powder	Gated Pooling, CNN	Automated Image Sorting System	3360 Image Samples Across Seven Categories (Pure Turmeric Powder, Chickpea Flour, and Their Mixtures)	99.70%	Laboratory Phase
[68]	Detection of Adulteration in Turmeric Powder	Gated Pooling, CNN	Image Acquisition System	6240 Image Samples Across 13 Categories (Pure Turmeric Powder, Chickpea Flour, and Their Mixtures)	97.50%	Laboratory Phase
[69]	Qualitative and Quantitative Identification of Adulterated Marine Fishmeal	CNN, NIR-HSI	Near-Infrared Hyperspectral Imaging System	53 MFM Samples, 10 HFM Samples, and 16 FBM Samples	99.37%	Laboratory Phase
[70]	Detection of Adulteration and Quality of Saffron	LA Strategy, CNN	ALE-L121 Camera	1869 Image Samples Across Six Categories	99.50%	Laboratory Phase

**Table 5 foods-14-00247-t005:** Applications of CNN in food classification and recognition.

References	Objective	Analyses	Equipment	Dataset	Accuracy	Technology Maturity
[71]	Development of a Nutrition Analysis System for Automated Food Classification	Mask R-CNN	Standard Camera	Ville Cafe Dataset (16 Classes, 35,842 Images) and Food-256 Dataset	99.86%	Laboratory Phase
[51]	Classification of Food Images to Identify Unknown Foods and Their Calorie Content	CNN	None	FOOD-101 Dataset	86.85%	Laboratory Phase
[72]	Rapid and Low-Cost Classification of Squid Species and Assessment of Freshness	Faster R-CNN	iPhone X Camera	600 Squid Sample Images	85.70%	Laboratory Phase
[73]	Classification of Barley Varieties	CNN	Flatbed Scanner	Images of 11 Barley Varieties	Above 93%	Laboratory Phase
[74]	Classification of Hazelnut Varieties	CNN with Pretrained Models	14.1-Megapixel Camera	4250 Hazelnut Image Samples	98.63%	Practical Application Phase

**Table 6 foods-14-00247-t006:** Applications of CNN in food damage detection.

References	Objective	Analyses	Equipment	Dataset	Accuracy	Technology Maturity
[75]	Detection of Mechanical Damage to Sugar Beets During Harvesting	YOLO v4, Faster R-CNN	High-Speed Camera	3425 Images of Sugar Beets	94%	Practical Application Phase
[76]	Detection of Damage in Polished Rice Grains	DCNN	Sony Camera	8048 High-Magnification Images of Damaged Rice	98.37%	Laboratory Phase
[77]	Detection of Surface Damage on Apples	RSWL, CNN	Red Dot Laser Emitter and Spectrometer	Spectral Images of 1125 Sets of Fuji Apple Samples	93%	Laboratory Phase
[78]	Detection of Early Frost Damage in Oranges	Vis/NIR, 1D-CNN	Online Spectral Measurement System	Transmission Spectral Images of 114 Oranges	91.96%	Practical Application Phase
[79]	Detection of Insect Damage in Maize Seeds	1D-CNN-BiLSTM, SVM	Near-Infrared Hyperspectral Imaging System	Spectral Images of 240 Maize Seed Samples	96%	Laboratory Phase

**Table 7 foods-14-00247-t007:** Application of RNNs in public sentiment analysis.

References	Objective	Analyses	Dataset	Accuracy	Technology Maturity
[111]	Detection of Food Safety Incidents on Social Media	LSTM	A Food Safety-Related Dataset Collected from Twitter, Consisting of 2418 Manually Annotated Tweets	F1 Score Outperforms Baseline Methods	Laboratory Phase
[112]	Development of a Rapid Early Warning System to Identify the Correlation Between Online Food Risks and Public Opinion Data	LSTM, Group Learning	A Dataset Containing Approximately 9000 and 3000 Sentences, Labeled as Positive, Negative, and Neutral	Good Performance and Effectiveness	Laboratory Phase
[113]	Collection and Analysis of Public Opinion on Food Safety, and Development of a Real-Time Sentiment Analysis Platform	MLP, LSTM	Over 3100 Public Data Sources, Including News Websites, E-Newspapers, and Forums	F1 Score: 97.37%	Practical Application Phase
[114]	Evaluation of the Influence Level of Public Opinion on Food Safety and Addressing the Problem of Imbalanced Samples	SVM, LSTM, BERT	Real Public Opinion Data from the Food Safety Department of China Customs, Including 21,065 Samples	Accuracy: 85.3%	Practical Application Phase

**Table 8 foods-14-00247-t008:** Application of RNNs in supply chain demand forecasting.

References	Objective	Analyses	Dataset	Accuracy	Technology Maturity
[115]	Prediction of Order Quantities in the Food Supply Chain	RFR, GBR, XGB, LSTM, BiLSTM	The “Food Demand Forecasting” Dataset Released by Genpact	Mean Error: 6.56%	Practical Application Phase
[116]	Comparison of SARIMA and LSTM for Demand Forecasting in the Retail Supply Chain	SARIMA, LSTM	Actual Retail Sales Data Over a 37-Month Period from an Austrian Retailer	LSTM is More Suitable	Practical Application Phase

**Table 9 foods-14-00247-t009:** Application of RNNs in environmental condition prediction.

References	Objective	Analyses	Dataset	Accuracy	Technology Maturity
[117]	Prediction of Food Temperature	ARIMA, RNN, LSTM	Three Months of Food Temperature Records	LSTM Accuracy is Higher Than Other Methods	Laboratory Phase
[118]	Prediction of Temperature and Humidity in Grain Storage Facilities	RNN-LSTM	Storage Grain and Food Condition Data for Short-Term or Small Datasets	LSTM Accuracy is Higher Than Other Methods	Laboratory Phase
[119]	Prediction of Temperature and Humidity Variation Trends in Agricultural Cold Chain Transportation Environments	k-means++, LSTM	Approximately 1500 Temperature and Humidity Data Points Collected Along a Transportation Route in China	Temperature: RMSE = 0.5707, Humidity: RMSE = 1.6015	Practical Application Phase
[120]	Prediction of Evapotranspiration and Relative Humidity in Tomato Greenhouses	LSTM	Time Series Data from Environmental Sensors and Actuators	Humidity Prediction: RMSE = 2.87	Laboratory Phase
[121]	Prediction of Soil Moisture in Soybean Cultivation	RNN-LSTM	Soil Moisture and Temperature Sensor Data for One Quarter in 2020	R^2^ = 0.999	Practical Application Phase

**Table 10 foods-14-00247-t010:** Application of RNNs in risk early warning and analysis.

References	Objective	Analyses	Dataset	Accuracy	Technology Maturity
[122]	Propose a Food Safety Risk Early Warning Method Considering Environmental and Nutritional Factors	LSTM, AHP-SP	1236 Sets of Pasteurized Milk Data from a Food Testing Agency in China	LSTM Accuracy is Higher Than Other Methods	Practical Application Phase
[123]	Develop an IAPSO-LSTM-Based Food Safety Risk Early Warning Model Specially Designed for Predicting Potential Contamination Risks in Food	LSTM, AHP-SP	Compound Seasoning Testing Dataset Containing 849 Samples for Model Training and 253 Samples for Risk Value Validation	R^2^ = 0.998	Practical Application Phase
[22]	Predict Food Safety Compliance Rates by Establishing a Prediction Model Based on the RL-ALSTM Framework	LSTM, RL	Publicly and Internally Curated Dataset with Over 500,000 Food-Grade Sampling Records	RMSE = 0.0316	Laboratory Phase
[124]	Predict the Safety Risk Levels of Benzo [a] pyrene Residues in Edible Oil for Forecasting Food Safety Risks in Edible Oil	LSTM, XGBoost	National Sampling Data from 20 Provinces and Their Sub-Regions in China in 2019	RMSE = 0.0018	Laboratory Phase
[125]	Design a Spatiotemporal Risk Prediction Model for Forecasting Foodborne Disease Risks in Different Regions	MS-LSTM, STA	Foodborne Disease Dataset from Multiple Provinces in China from 2015 to 2019	F1 Score: 0.67–0.82	Practical Application Phase

## Data Availability

No new data were created or analyzed in this study. Data sharing is not applicable to this article.

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
