# Peer review of "Application of Convolutional Neural Networks and Recurrent Neural Networks in Food Safety"

_foods, 2025, doi:10.3390/foods14020247_

Round 1
Reviewer 1 Report
Comments and Suggestions for Authors
This review paper gives a clear overview of how CNNs and RNNs, especially LSTMs, are used in food safety detection and risk prediction. The paper also discusses how these models are combined with technologies like IoT, blockchain, and federated learning in a practical and engaging way. Some issues should be clarified.
- What are the key benefits of combining HSI with CNNs for food safety detection, and how does it compare to using traditional RGB images?
- How does combining CNNs with ViT help address the challenges of analyzing complex food images, and what are the potential limitations of this approach?
- What are the main obstacles preventing the widespread industrial use of CNN models in food safety, particularly in real-time production settings?
- How do LSTM-based models outperform traditional statistical approaches in time-series analysis for predicting food safety risks?
- What challenges does using LSTM pose in resource-limited environments, and how could attention mechanisms or graph neural networks help address these?
- How do CNNs compare to traditional neural networks when processing image data specific to food safety applications?
- What are the key differences between RNNs and traditional statistical methods in managing multivariate time-series data for food safety risk predictions?
- How can CNNs and RNNs be effectively integrated with IoT and blockchain technologies to improve food safety detection and risk prediction?
- What potential issues could arise when integrating federated learning with CNNs and RNNs in food safety applications?
- What advancements in hardware and algorithm optimization are needed to boost the computational efficiency and scalability of CNN-ViT and LSTM models for practical applications?
- How does the limited size and diversity of datasets affect the generalization of CNN and RNN models in food safety contexts?
Reviewer 2 Report
Comments and Suggestions for Authors
This paper focuses on the potential of convolutional neural networks and recurrent neural networks, in food safety detection and risk prediction. Since the mentioned deep learning techniques have been applied recently and attracted the attention of researchers, the topic can be interesting for engineers and researchers who work on AI-driven systems in food engineering. However, the paper encountered some challenges. The structure and organization of the paper need revision. Much redundant and obvious information throughout the text was observed hence author should avoid them and summarize some parts. In addition, although the application of RNN and CNN in food safety was described comprehensively, lthe ack of adequate discussion was tangible.
L15- what is the meaning of ‘’ field of food safety, particularly in food safety detection and risk prediction’’. I suggest to revise the sentence.
L44- Please merge 1.1 and 1.2. as one section with the introduction.
L45-48 It seems there is a mistake in the citation, please double-check it.
L65-80 Although the traditional ML models in some cases aren’t useful for food safety prediction, their accuracy in most cases is acceptable. Thus, I suggest revising this paragraph and just highlighting some disadvantages of traditional ML models in some cases.
L171- Figure 1 illustrates the basic structure of CNNs and its process of automatically extracting food shape features for food recognition. Please describe its details.
L192- While the structure of CNN were described comprehensively, the lack of enough discussion was tangible.
L407- Figure 4 presents deep learning techniques applied in food safety that I believe that its place is not appropriate. The figure is suitable for the introduction section rather than the Food Safety Risk Prediction Using RNN section.
L469- In the text, just some literature was described and the current gap was not mentioned. Authors should compare the weaknesses and superiorities of applied models and discuss them.
L613- In the paper, the ‘’Challenges of Model Interpretability’’ couldn’t describe the current challenges and limitations of the RNN and CNN completely. Thus, I suggest considering another section and explaining the current limitations and challenges of RNN and CNN in food safety.
L724- The author should highlight the future prospect of application of CNN and RNN in food safety. This section can be considered as a separate part or authors can integrate this section with the conclusion.
Comments on the Quality of English LanguageThe English could be improved to more clearly express the research.
Reviewer 3 Report
Comments and Suggestions for Authors
The paper provides a review of analytical prediction models for food safety. I had some general comments I thought.
1. I would recommend avoiding using acronyms in the title. As they could coincide with other more popular, unrelated, acronyms. If readers are unaware of these, they may not understand what the paper is about.
2. Line 67. It is useful to highlight small datasets. However, it makes less sense when reading the following sentence, which includes decision trees and support vector machines. These both rely on big data. I think this could be reviewed to ensure the sentences are consistent.
3. Line 74. This is not only, or is at least, a partial description of Bayesian statistics. This approach uses priors that up updated with new data. Therefore, the method is effective when historical data has been accrued.
4. Line 78. This is not correct. These models can be used to handle non-linear relationships by using Kernels and are to accurately classify or predict outcomes. They may be able to identify patterns, but I am uncertain whether they outperform pattern analyses such as PCA, LDA, or cluster analysis.
5. Generally, I am somewhat confused about the distinction between traditional and deep learning methods. The traditional methods also contain machine learning methods, which are more similar to deep learning methods than other methods (e.g. time series and regression analysis.) The lack of distinction seems to create some assertions about
6. The aim of the review may need revision. It can be more explicit about the application of CNN and RNN for the predictor of food safety and risk detection.
7. Should some images be used to understand better how food detection works? For example, which situations are needed to ensure models correctly predict unsafe and safe foods?
8. Tables 1 and 2 are a comprehensive summary of the review. However, the information is difficult to read in this format. I would recommend dividing the table by the first columns and then removing this column. The other columns can then be widened. The reference can also be moved to the first column and the second column (‘Methods’) renamed to ‘Analyses.’
9. Figure 5. Although it is helpful to highlight the challenges, the two examples provided are very general and are somewhat the same. The Black-box model is something nearly all researchers encounter using these types of models. Maybe the second point (‘Strict Industry Standards’) is more relevant to highlight here.
10. IoT Technology. The drawback of IoT seems quite general on line 563. If this is the case, whichever sensing technology is used will also have this limitation as it has to do with data measurement, not data collection. IoT-enabled technologies handle data being collected. A more pressing issue could be how the data is collected, managed and stored as servers are needed which could use a lot of energy. Also, depending on the scale, how many sensors require IoT to collect data?
Round 2
Reviewer 1 Report
Comments and Suggestions for Authors
This revised version seems good. However, some issues should be clarified.
How do CNNs outperform fully connected neural networks in extracting spatial features from food-related images?
How do model interpretability limitations affect the adoption of CNNs and RNNs in regulatory environments?
Reviewer 2 Report
Comments and Suggestions for Authors
I appreciate the author's efforts in the revision process.
Reviewer 3 Report
Comments and Suggestions for Authors
Thank you for providing detailed amendments to your work. I have no additional comments or suggestions to at to this paper.
